Resource

# Systematic identification of ALK substrates by integrated phosphoproteome and interactome analysis

Jun Adachi[1,2,3] , Akemi Kakudo[1,2], Yoko Takada[1,2], Junko Isoyama[1,2], Narumi Ikemoto[1,2], Yuichi Abe[1,2], Ryohei Narumi[1,2], Satoshi Muraoka[1,2], Daigo Gunji[2,4], Yasuhiro Hara[2], Ryohei Katayama[5,6], Takeshi Tomonaga[1,2]

The sensitivity of phosphorylation site identification by mass spectrometry has improved markedly. However, the lack of kinase–substrate relationship (KSR) data hinders the improvement of the range and accuracy of kinase activity prediction. In this study, we aimed to develop a method for acquiring systematic KSR data on anaplastic lymphoma kinase (ALK) using mass spectrometry and to apply this method to the prediction of kinase activity. Thirty-seven ALK substrate candidates, including 34 phosphorylation sites not annotated in the PhosphoSitePlus database, were identified by integrated analysis of the phosphoproteome and crosslinking interactome of HEK 293 cells with doxycycline-induced ALK overexpression. Furthermore, KSRs of ALK were validated by an in vitro kinase assay. Finally, using phosphoproteomic data from ALK mutant cell lines and patient-derived cells treated with ALK inhibitors, we found that the prediction of ALK activity was improved when the KSRs identified in this study were used instead of the public KSR dataset. Our approach is applicable to other kinases, and future identification of KSRs will facilitate more accurate estimations of kinase activity and elucidation of phosphorylation signals.

## Introduction

Protein phosphorylation is a major regulator of intracellular signalling. Its dynamic plasticity governs a cell's response to its environment. Dysregulation of phosphorylation signalling is often implicated in pathogenesis and is a major therapeutic target in diverse diseases, including cancer and neurodegenerative disorders. Technological advances in mass spectrometry (MS) have enabled us to quantitate tens of thousands of phosphorylation sites from a variety of samples, even small samples such as biopsy specimens (Abe et al, 2020; Satpathy et al, 2020). Various informatics methods have been developed to extract biological insights, such as the activity of kinases and signalling pathways, from large-scale phosphorylation data (Linding et al, 2007; Casado et al, 2013; Krug et al, 2019). In these analyses, only a small fraction (often less than 5%) of the quantified phosphorylation sites were used, mainly because much less information is available on known kinase–substrate relationships (KSRs) than on phosphorylation sites quantified by MS (Needham et al, 2019). Various strategies have been used to obtain high-quality KSR data on a large scale; these approaches include informatics approaches (Invergo et al, 2020; Nováček et al, 2020), approaches using in vitro kinase assays (Knebel et al, 2001; Newman et al, 2013; Sugiyama et al, 2019), chemical proteomics approaches using kinase inhibitors in vivo (Hijazi et al, 2020; Watson et al, 2020), and biochemical and genetic approaches combining proximity-dependent biotinylation (BioID)-based interactome and phosphoproteome analyses (Cutler et al, 2020; Niinae et al, 2021). The informatics approach relies on public phosphoproteome data or KSR data as input data; thus, predicting new substrate candidates for a kinase with limited KSR data is difficult by this approach. Because the in vitro kinase assay does not reflect aspects of the intracellular environment, such as localization and complex formation, its data contain false-positive hits of sites that are not phosphorylated in vivo. To remove false-positive hits, these data must be combined with additional data, such as kinase perturbation data (Xue et al, 2012; Imamura et al, 2017). The chemical proteomics approach makes it relatively easy to obtain in vivo data. However, this approach cannot be applied to all kinases because it requires a specific inhibitor for the target kinase. On the other hand, establishing a genetically engineered cell line via biochemical and genetic approaches requires time; however, these approaches are straightforward and have the potential to expand the experimental scale to the kinome level. In this study, we used a combined biochemical and genetic approach in which we established doxycycline (Dox)-induced anaplastic lymphoma kinase (ALK)-overexpressing HEK 293 cells and analysed the formaldehyde crosslinking interactome and

[1]Laboratory of Proteomics for Drug Discovery, Center for Drug Design Research, National Institutes of Biomedical Innovation, Health and Nutrition, Osaka, Japan [2]Laboratory of Proteome Research, National Institutes of Biomedical Innovation, Health and Nutrition, Osaka, Japan [3]Laboratory of Proteomics and Drug Discovery, Graduate School of Pharmaceutical Sciences, Kyoto University, Kyoto, Japan [4]Department of Surgery, Kyoto University Graduate School of Medicine, Kyoto, Japan [5]Division of Experimental Chemotherapy, Cancer Chemotherapy Center, Japanese Foundation for Cancer Research, Tokyo, Japan [6]Department of Computational Biology and Medical Sciences, Graduate School of Frontier Sciences, The University of Tokyo, Tokyo, Japan

Correspondence: jun_adachi@nibiohn.go.jp

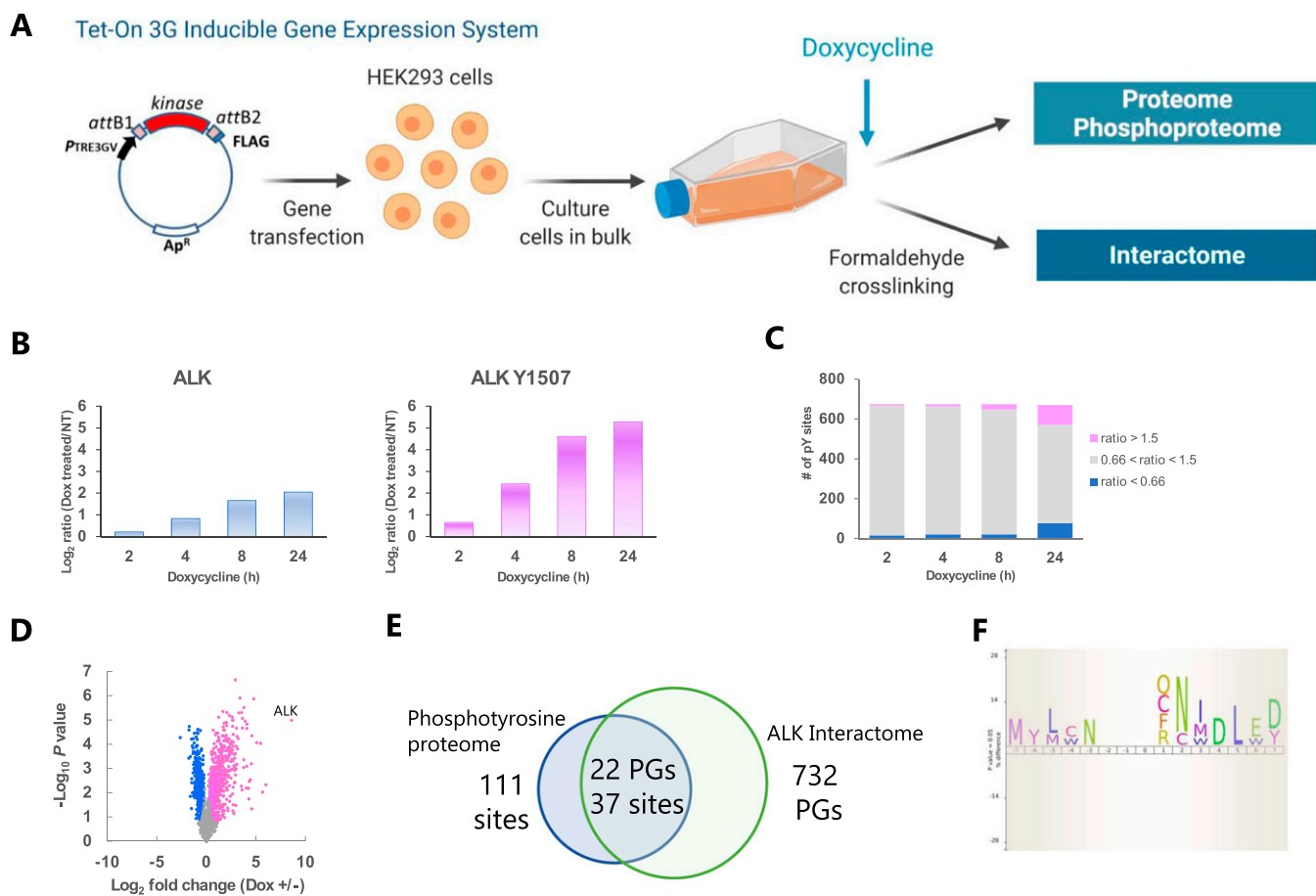

**Figure 1. Identification of anaplastic lymphoma kinase (ALK) substrate candidates by phosphoproteome and interactome analysis of Dox-inducible HEK 293 cells.**
**(A)** Overview of the Dox-inducible ALK gene expression system and the phosphoproteome and interactome analysis methods. **(B)** Time-course data of ALK protein expression and phosphorylation. **(C)** Time-course data of phosphotyrosine proteome analysis. The pink and blue bars represent phosphotyrosine sites at which phosphorylation was altered by more than 1.5-fold (i.e., up-regulated) or less than 0.66-fold (i.e., down-regulated), respectively, compared with that in non–Dox-treated cells. **(D)** Volcano plot of interactome analysis data. The pink and blue dots represent significant precipitated proteins ($q < 0.05$). **(E)** Venn diagram showing the overlap between the phosphotyrosine proteome and interactome data. **(F)** Sequence motif analysis by iceLogo of the amino acid residues between positions ±7 adjacent to the phosphorylation sites of ALK substrate candidates. The significance threshold was set to 0.05.
Source data are available for this figure.

time-course of the phosphotyrosine (pY) proteome to obtain KSR data for ALK. We validated the usefulness of this approach in predicting ALK activity using phosphoproteomic data from ALK-mutated cultured cells, patient-derived cells and clinical samples.

## Results

### Identification of ALK substrate candidates by combined phosphoproteome and interactome analysis

To systematically identify ALK substrates, we established cells with Dox-inducible *ALK* expression based on a Tet-On 3G–inducible gene expression system (Fig 1A). The protein expression level of ALK was increased 4.1-fold, and the phosphorylation level of ALK Tyr[1507] was increased 38.9-fold by the addition of Dox (50 ng/ml) for 24 h (Fig 1B). Using our cells with Dox-inducible *ALK* expression, we

performed time-course analysis of the phosphotyrosine proteome. We quantified 655 phosphotyrosine sites and identified 111 phosphotyrosine sites whose phosphorylation was up-regulated more than 1.5-fold at 2, 4, 8, or 24 h after Dox induction (Fig 1C). We also performed interactome analysis using formaldehyde crosslinking. After crosslinking, Dox-induced cells (Dox+) and control cells (Dox−) were lysed, and ALK-interacting proteins were immunoprecipitated with an anti-flag antibody and quantified by label-free quantitation (LFQ). We identified 732 protein groups that were significantly precipitated ($q < 0.05$) (Fig 1D). Finally, we selected 37 phosphotyrosine sites (among 22 protein groups) that overlapped between the up-regulated phosphoproteome and ALK interactome as candidate ALK substrates (Fig 1E and Table 1). Sequence motif analysis of the ALK substrate candidates revealed that the SH2 domain-binding motif pYXN (Schlessinger & Lemmon, 2003) was enriched (Fig 1F). In addition to well-known ALK substrates, such as the adaptor protein SHC1 (Tyr[427]) and the protein tyrosine phosphatase PTPN11 (Tyr[279], Tyr[542], and Tyr[580]), ANXA2 (Tyr[30]), APLP2

**Table 1. Identified phosphorylation sites of ALK substrate candidates.**

| Proteins | Positions within proteins | Protein names | Gene names |
|---|---|---|---|
| Q9UM73 | 1,096 | ALK tyrosine kinase receptor | ALK |
| Q9UM73 | 1,507 | ALK tyrosine kinase receptor | ALK |
| Q9UM73 | 1,586 | ALK tyrosine kinase receptor | ALK |
| Q9UM73; P29376 | 1,278; 672 | ALK tyrosine kinase receptor; leukocyte tyrosine kinase receptor | ALK; LTK |
| Q9UM73; P29376 | 1,282; 676 | ALK tyrosine kinase receptor; leukocyte tyrosine kinase receptor | ALK; LTK |
| Q9UM73; P29376 | 1,283; 677 | ALK tyrosine kinase receptor; leukocyte tyrosine kinase receptor | ALK; LTK |
| P07355; A6NMY6 | 30; 30 | Annexin A2; putative Annexin A2-like protein | ANXA2; ANXA2P2 |
| Q06481 | 750 | Amyloid-like protein 2 | APLP2 |
| P09543 | 110 | 2′,3′-cyclic-nucleotide 3′-phosphodiesterase | CNP |
| Q7L576; Q96F07 | 1,054; 1,078 | Cytoplasmic FMR1–interacting protein 1; cytoplasmic FMR1–interacting protein 2 | CYFIP1; CYFIP2 |
| P50402 | 161 | Emerin | EMD |
| Q9BSJ8 | 359 | Extended synaptotagmin-1 | ESYT1 |
| Q9BSJ8 | 822 | Extended synaptotagmin-1 | ESYT1 |
| A0FGR8 | 824 | Extended synaptotagmin-2 | ESYT2 |
| Q96CS3 | 79 | FAS-associated factor 2 | FAF2 |
| O75955 | 216 | Flotillin-1 | FLOT1 |
| O75955 | 238 | Flotillin-1 | FLOT1 |
| O60547 | 323 | GDP-mannose 4,6 dehydratase | GMDS |
| O14654 | 151 | Insulin receptor substrate 4 | IRS4 |
| O14654 | 291 | Insulin receptor substrate 4 | IRS4 |
| O14654 | 336 | Insulin receptor substrate 4 | IRS4 |
| O14654 | 615 | Insulin receptor substrate 4 | IRS4 |
| O14654 | 656 | Insulin receptor substrate 4 | IRS4 |
| O14654 | 921 | Insulin receptor substrate 4 | IRS4 |
| Q14847 | 57 | LIM and SH3 domain protein 1 | LASP1 |
| P07948 | 363 | Tyrosine-protein kinase Lyn | LYN |
| O95819 | 467 | Mitogen-activated protein kinase kinase kinase kinase 4 | MAP4K4 |
| Q9Y237 | 122 | Peptidyl-prolyl cis–trans isomerase NIMA-interacting 4 | PIN4 |
| Q06124 | 279 | Tyrosine-protein phosphatase non-receptor type 11 | PTPN11 |
| Q06124 | 546 | Tyrosine-protein phosphatase non-receptor type 11 | PTPN11 |
| Q06124 | 584 | Tyrosine-protein phosphatase non-receptor type 11 | PTPN11 |
| P10586-2; P10586 | 1,612; 1,621 | Receptor-type tyrosine-protein phosphatase F | PTPRF |
| P10586; P23468; Q13332 | 1,381; 1,386; 1,422 | Receptor-type tyrosine-protein phosphatase F; receptor-type tyrosine-protein phosphatase delta; receptor-type tyrosine-protein phosphatase S | PTPRF; PTPRD; PTPRS |
| P49023 | 88 | Paxillin | PXN |
| P29353 | 427 | SHC-transforming protein 1 | SHC1 |
| P84022; Q99717; Q15797; Q15796; O15198 | 88; 89; 88; 128; 92 | Mothers against decapentaplegic homolog 3; mothers against decapentaplegic homolog 9; mothers against decapentaplegic homolog 2; mothers against decapentaplegic homolog 5; mothers against decapentaplegic homolog 1 | SMAD3; SMAD9; SMAD2; SMAD5; SMAD1 |
| Q9BZV1 | 336 | UBX domain-containing protein 6 | UBXN6 |

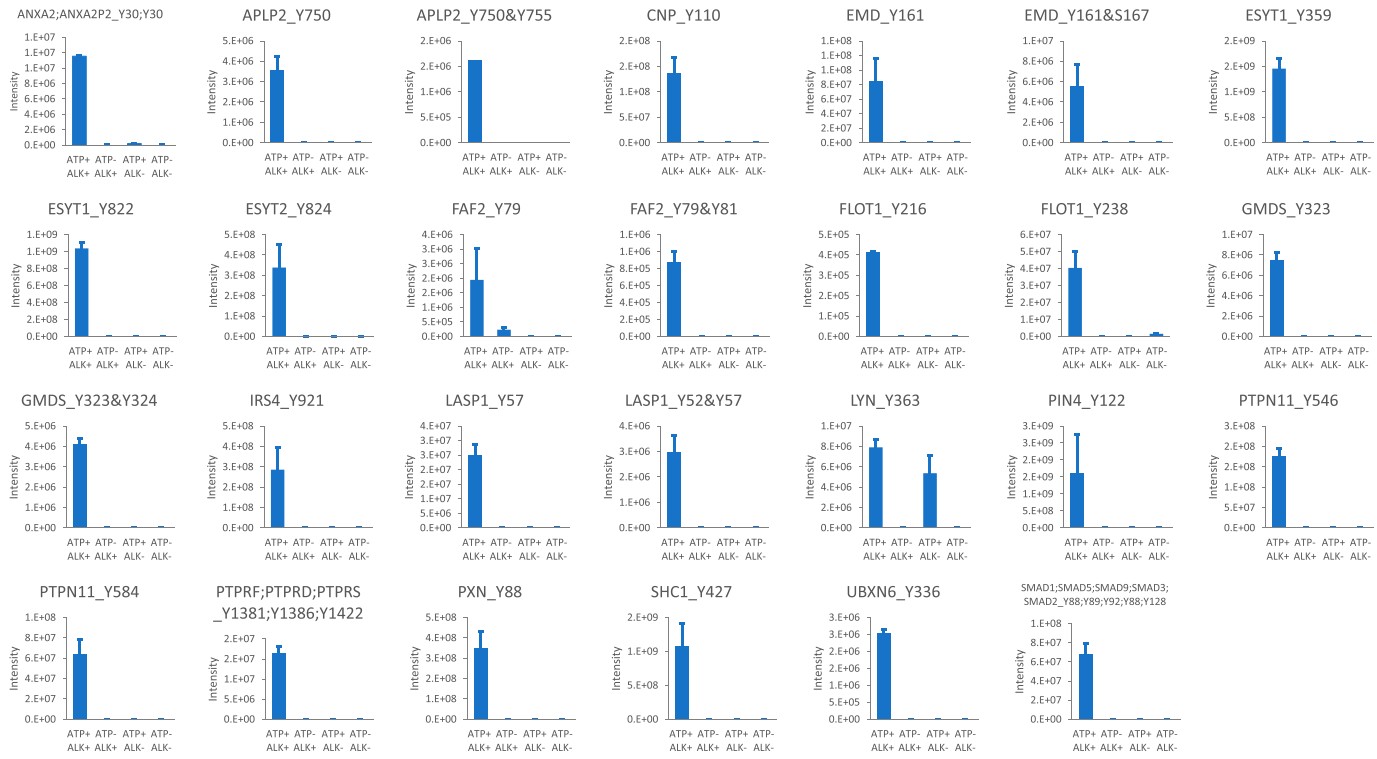

**Figure 2. In vitro kinase assay of anaplastic lymphoma kinase (ALK) substrate candidates.**
An in vitro kinase assay of 27 ALK substrate candidates was performed in the presence or absence of ATP and the presence or absence of ALK. The results are expressed as the mean ± SD values.
Source data are available for this figure.

(Tyr[750]), IRS4 (Tyr[921]), and PTPRF/PTPRD/PTPRS (Tyr[1381]/Tyr[1386]/Tyr[1422]) contain a pYXN motif.

### Confirmation of ALK–substrate relationships by an in vitro kinase assay

To assess the direct phosphorylation of ALK, we performed an in vitro kinase assay using 21 candidate ALK substrate proteins. As shown in Fig 2, we assessed phosphorylation at 27 phosphotyrosine sites and confirmed that all analysed sites except Lyn (Tyr[363]) were significantly phosphorylated in the presence of ATP and ALK. This finding suggested that Lyn (Tyr[363]) is phosphorylated in the absence of ALK by autophosphorylation.

### Validation of ALK–substrate relationships by phosphoproteome analysis of neuroblastoma cell lines

*ALK* amplification or mutation was identified in ~14% of neuroblastomas (NBs), the most common extracranial childhood tumour (Chen et al, 2008; George et al, 2008; Janoueix-Lerosey et al, 2008; Mosse et al, 2008). To examine the phosphorylation status of ALK substrate candidates identified in our study, the neuroblastoma cell lines NB-1 and KP-N-RT-BM-1, which harbour amplification of full-length *ALK* and the F1174L mutation, respectively, were selected as the ALK-activated cell lines. The IMR-32 cell line, without *ALK* gene mutation, was selected as a control cell line. These cells have a common genotype of *MYCN* (amplified) and *TP53* (wild-type)

(Wang et al, 2017). We performed phosphoproteome analysis, and 25 phosphopeptides of ALK substrate candidates were quantified (Fig 3). The protein expression level of ALK was not increased in KP-N-RT-BM-1 cells but was increased by 3.9-fold in NB-1 cells compared to IMR-32 cells. Five phosphopeptides of ALK were quantified; these phosphopeptides were increased 0.9- to 3.1-fold in KP-N-RT-BM-1 cells and 7.6- to 28.5-fold in NB-1 cells. Among the 25 phosphopeptides of ALK substrates, 19 were significantly increased in KP-N-RT-BM-1 cells, and 21 were significantly increased in NB-1 cells (*P* < 0.05). All phosphopeptides except for EMD Tyr[161], Tyr[161], and Ser[171] MAP4K4 Tyr[476] were up-regulated in either KP-N-RT-BM-1 or NB-1 cells.

We calculated the log$_2$-fold changes in phosphopeptides in KP-N-RT-BM-1 versus IMR-32 cells and in NB-1 versus IMR-32 cells and used these data for kinase activity prediction with PTM Signature Enrichment Analysis (PTM-SEA) (Krug et al, 2019). We compared default KSR data (ptm.sig.db.all.flanking.human.v1.9.0) and customized KSR data to which our ALK substrate candidates (37 phosphotyrosine sites) were added. As shown in Fig 4, the normalized enrichment score (NES) was increased 3.15–7.705 in KP-N-RT-BM-1 cells and 3.46–9.52 in NB-1 cells by changing the KSR dataset from the default setting to the custom setting. The adjusted *P*-value was also greater than 0.05 for KP-N-RT-BM-1 cells with the default KSR dataset but improved to less than 0.01 with the custom KSR dataset. We used PTM-SEA to analyse the phosphoproteome data of each neuroblastoma cell line. Predicted kinase activities were visualized in a kinome map (Fig 5) (Metz et al, 2018). Compared

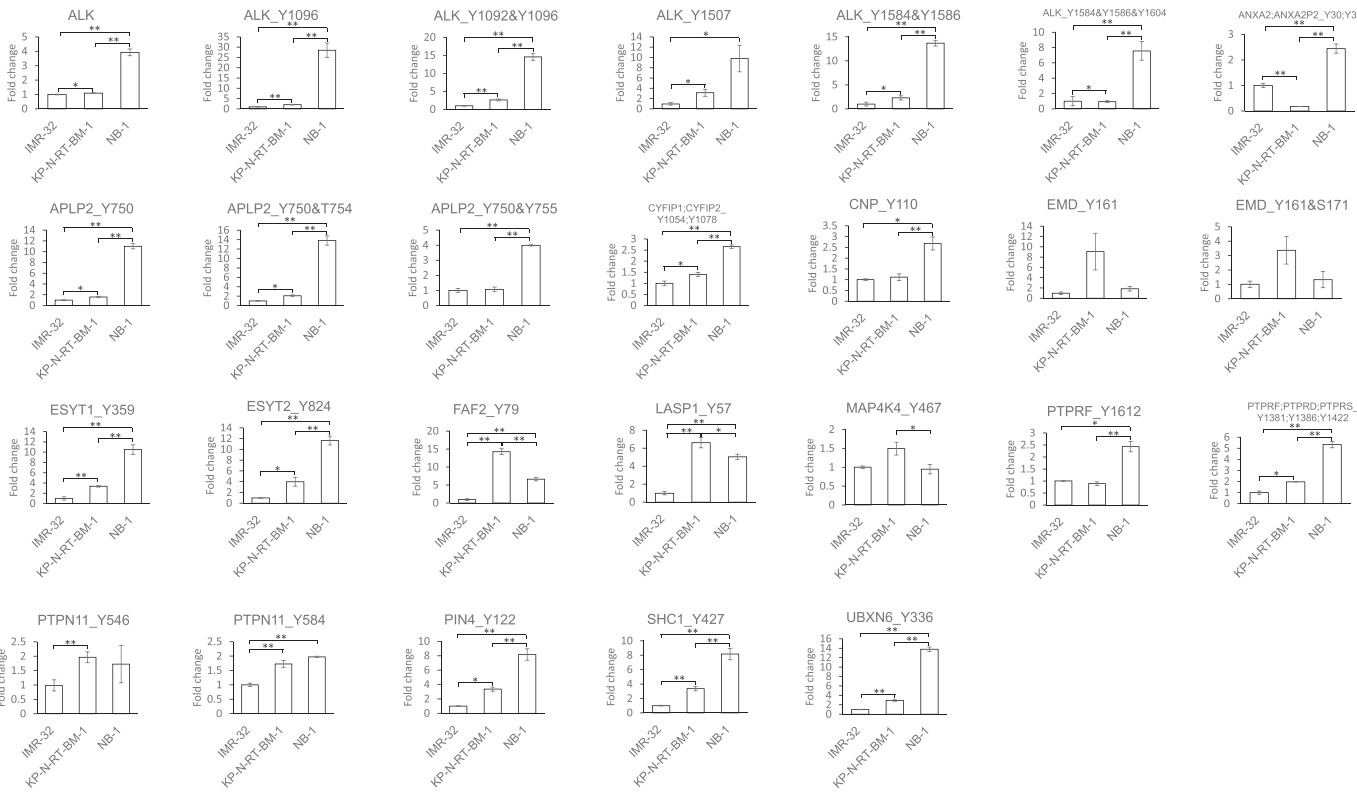

**Figure 3. Phosphorylation level of anaplastic lymphoma kinase substrate candidates in neuroblastoma cell lines.**
A fold changes in the phosphorylation sites of anaplastic lymphoma kinase substrate candidates. The phosphorylation levels were normalized to those in IMR-32 cells, which were set as 1. The asterisks indicate a significant difference by Welch's test (*$P < 0.05$, **$P < 0.01$).
Source data are available for this figure.

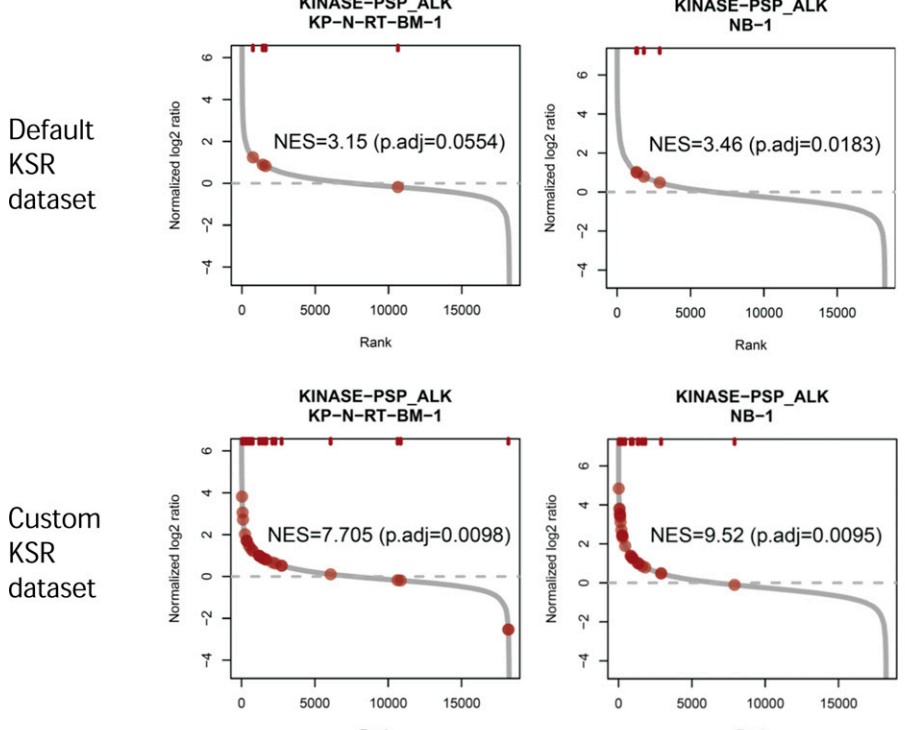

**Figure 4. Rank plot of anaplastic lymphoma kinase substrate candidates in neuroblastoma cell lines.**
The median of the phosphoproteomic expression data of each cell (n = 3) was analysed by PTM-SEA. The dark red circles indicate substrates of anaplastic lymphoma kinase.
Source data are available for this figure.

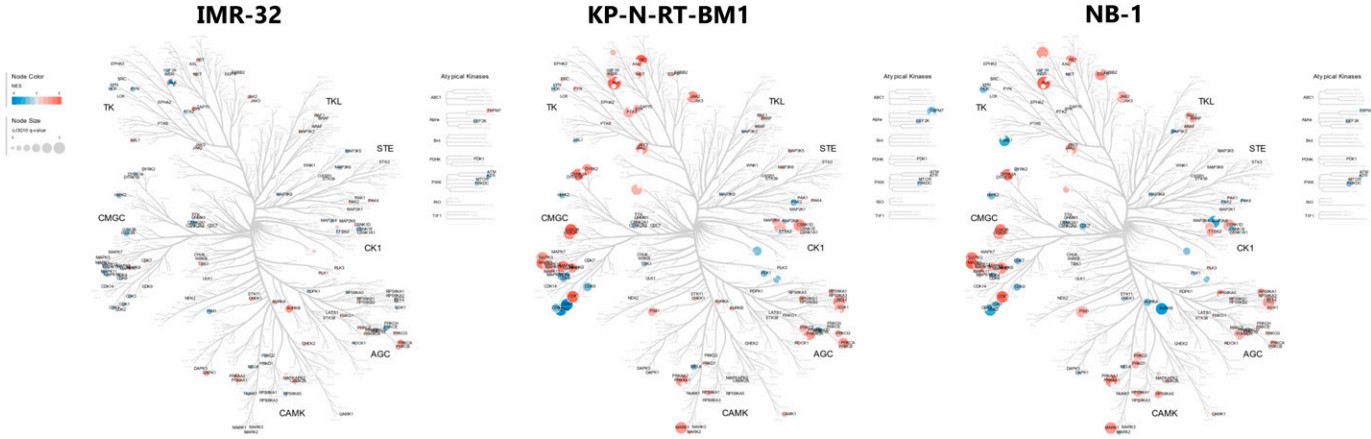

**Figure 5. Kinome activity profiles of neuroblastoma cell lines by PTM-SEA analysis.**
Kinome activity map of the neuroblastoma cell lines IMR-32 (*ALK* WT), KP-N-RT-BM1 (*ALK* F1174L), and NB-1 (*ALK* amp). The colour of each node represents the normalized enrichment score, and the size represents the $-\log_{10}$ *q*-value.
Source data are available for this figure.

with IMR-32 cells, KP-N-RT-BM-1 and NB-1 cells show a very similar pattern of kinome activity, although ALK is activated by *ALK* mutation and *ALK* amplification. In particular, the RAF-MAPKK-MAPK axis, AKT-GSK-3 axis, and CDK5 were commonly activated in KP-N-RT-BM-1 and NB-1 cells.

Furthermore, we examined the effect of an ALK inhibitor on ALK activity prediction using previously published deep phosphoproteome data obtained from NB-1 cells treated with the ALK inhibitors crizotinib, TAE684, and LDK378 (Emdal et al, 2018). As shown in Fig S1, inhibition of ALK activity by all the inhibitors was significantly predicted using the custom KSR dataset (adjusted *P*-value < 0.05) but not with the default KSR dataset.

### Application of ALK–substrate relationship information for predicting EML4-ALK activity

ALK activation in cancer is caused by ALK fusion proteins as well as ALK overexpression (Hallberg & Palmer, 2013). ALK fusion proteins are found in a wide range of cancer types. Crizotinib, the first clinically approved drug to target ALK, is a tyrosine kinase inhibitor that was approved for use in echinoderm microtubule-associated protein-like 4 *EML4-ALK*-positive non-small-cell lung cancer (NSCLC). Thus, we tried to detect upregulation of ALK activity in *EML4-ALK*-positive cells using our recently published data (Mizuta et al, 2021). As shown in Fig 6A, we compared the phosphorylation status of ALK substrates and predicted ALK activity in an *EML4-ALK*-positive NSCLC cell line (H3122), EML4-ALK–positive NSCLC patient-derived cell line (JFCR-028-3), *EML4-ALK*-I1171N mutant NSCLC patient-derived cell line (MCC-003) and *ALK* WT NSCLC cell line (A549). With the custom KSR dataset, the NES values were increased in H3122, JFCR-028-3, and MCC-003 cells and decreased in A549 cells compared with the corresponding values obtained with the default KSR dataset. In other words, the custom KSR dataset improved our ability to detect the activation state of ALK. In addition, differential phosphoproteome data comparing cells treated with gilteritinib, a newly discovered ALK inhibitor, and cells treated

with DMSO were applied to predict ALK activity. As shown in Fig 6B, inhibition of ALK activity was predicted only in MCC-003 cells using the default KSR dataset (adjusted *P*-value < 0.01). In contrast, inhibition of ALK activity was predicted in JFCR-028-3 and MCC-003 cells with the custom KSR dataset. In H3122 cells, the adjusted *P*-value decreased from 0.43 to 0.0116, and the NES decreased from −1.519 to −7.509, indicating that the ability of PTM-SEA to predict EML4-ALK activity was improved with the custom KSR dataset.

Furthermore, to validate the results using clinical specimens, we extracted phosphoproteome data for tumour tissue and normal tissue adjacent to the tumour (NAT) in EML4-ALK–positive patients from Clinical Proteomic Tumor Analysis Consortium (CPTAC) lung adenocarcinoma data (Gillette et al, 2020). In all quantified samples, ALK Y1507 and GMDS Y323 or Y324 were up-regulated, and ANXA2 Y30 and PXN Y88 were down-regulated. The other four phosphotyrosine sites—APLP2 Tyr$^{682}$, PTPN11 Tyr$^{546}$ and Tyr$^{584}$, and SHC1 Tyr$^{427}$—exhibited both increases and decreases among the samples (Fig 7A and B). The heat map showed that most of the substrates tended to be up-regulated in three cases (C3N.00572, C3L.00442, and C3N.02422) but tended to be down-regulated in other cases (C3N.00578, C3N.02587, C3N.00552, and C3N.00550).

## Discussion

Because ALK was initially discovered and characterized in a rare type of lymphoma called anaplastic large-cell lymphoma (ALCL) as an NPM-ALK fusion protein (Morris et al, 1994), *ALK* mutation and overexpression have been found in many cancer types. With the remarkable development of phosphoproteomics, large-scale analyses of signals downstream of ALK have been reported (Borenäs et al, 2021), and the relevance of ALK activity to disease is becoming clearer. However, large-scale substrate identification has not yet been achieved, as direct substrate identification requires the confirmation of a direct interaction. At this stage, knowledge of ALK

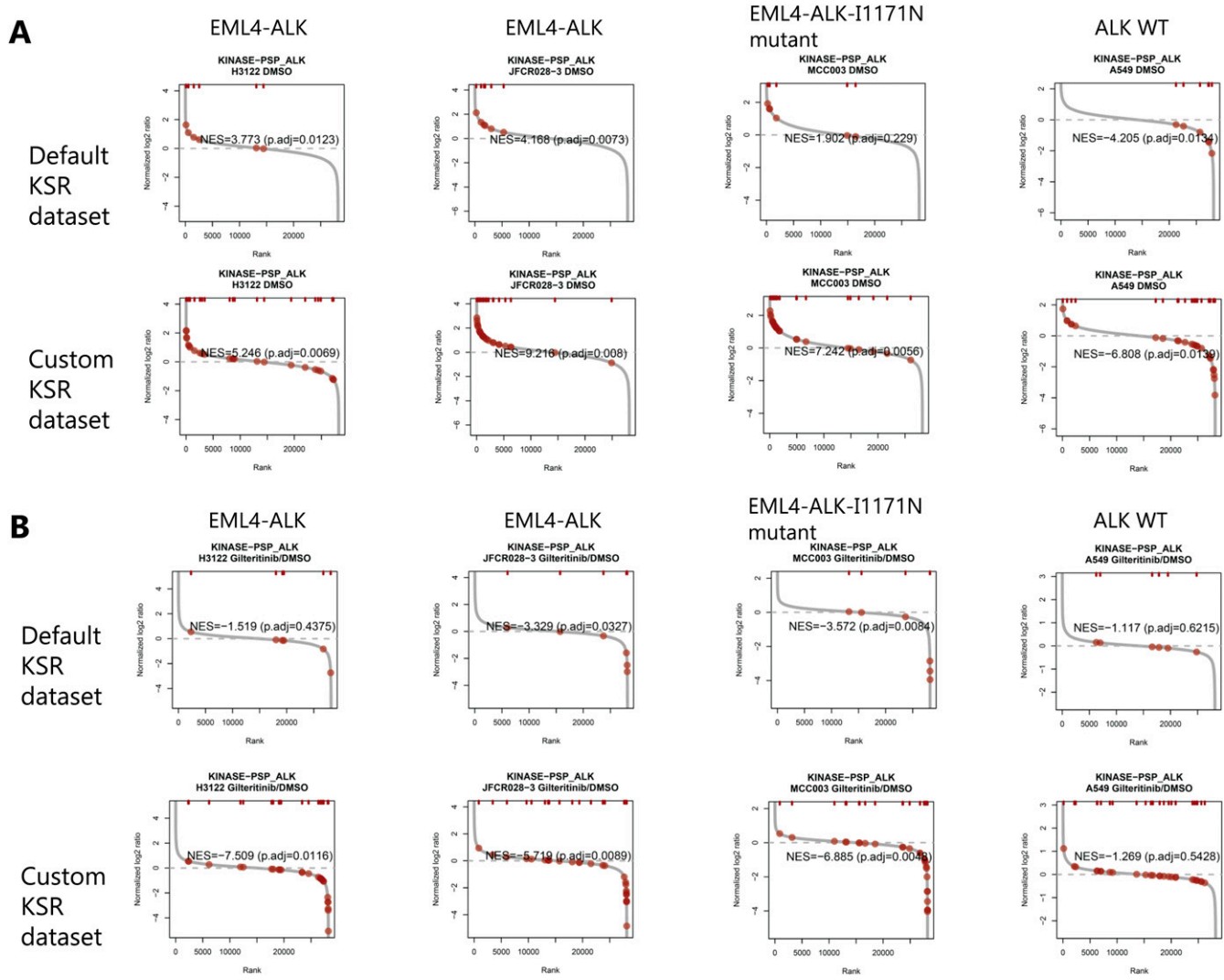

**Figure 6. Phosphorylation status of anaplastic lymphoma kinase substrate candidates as indicated by rank plots in NSCLC cells.**
**(A, B)** Rank plot of phosphoproteome data for DMSO-treated NSCLC cells (A) and rank plot of phosphoproteome data for NSCLC cells treated with gilteritinib (B). The average of the phosphoproteomic data of each cell (n = 3) was analysed by PTM-SEA. The dark red circles indicate substrates of anaplastic lymphoma kinase. Source data are available for this figure.

substrates is limited, and only 10 substrates in humans have been registered in the PhosphoSitePlus database. In this study, we identified 37 phosphotyrosine sites as ALK substrate candidates by integrated phosphoproteome and crosslinking interactome analysis of HEK 293 cells with Dox-inducible ALK overexpression. Phosphorylation sites of ALK substrate candidates were identified from based on MS/MS spectrum (Fig S2). In the future, it is desirable to confirm by other methods such as Western blotting using a site-specific antibody or a combination of immunoprecipitation and immunoblotting using an anti-phosphotyrosine antibody as shown in Fig S3. Among our ALK substrate candidates, ALK (Tyr$^{1278}$) and PTPN11 (Tyr$^{542}$ and Tyr$^{580}$) overlapped with KSRs annotated in the PhosphoSitePlus database. We also identified phosphorylation sites in the activation loop of ALK (Tyr$^{1282}$ and Tyr$^{1283}$) and other sites in ALK located in the intracellular region (Tyr$^{1096}$, Tyr$^{1507}$, and Tyr$^{1586}$).

Moreover, a well-known adaptor protein, SHC1 (Tyr$^{427}$), was identified as an ALK substrate.

In addition to these well-known ALK substrates, previously unreported substrate candidates were identified in this study, which contributed to the considerable improvement in ALK activity prediction (Figs 4, S1, and 6A and B). Some of these substrate candidates also formed a molecular association network (Fig S4). Recently, SMAD4 was reported to be phosphorylated at Tyr95 directly by ALK to elicit TGF-β gene transcription and tumour-suppressing responses (Zhang et al, 2019). We identified the corresponding site in SMAD1 (shared with SMAD2, SMAD3, SMAD5, and SMAD9) as a candidate ALK substrate. Thus, our data suggest that ALK is involved in TGF-β signalling through phosphorylation of a wide range of SMAD family proteins.

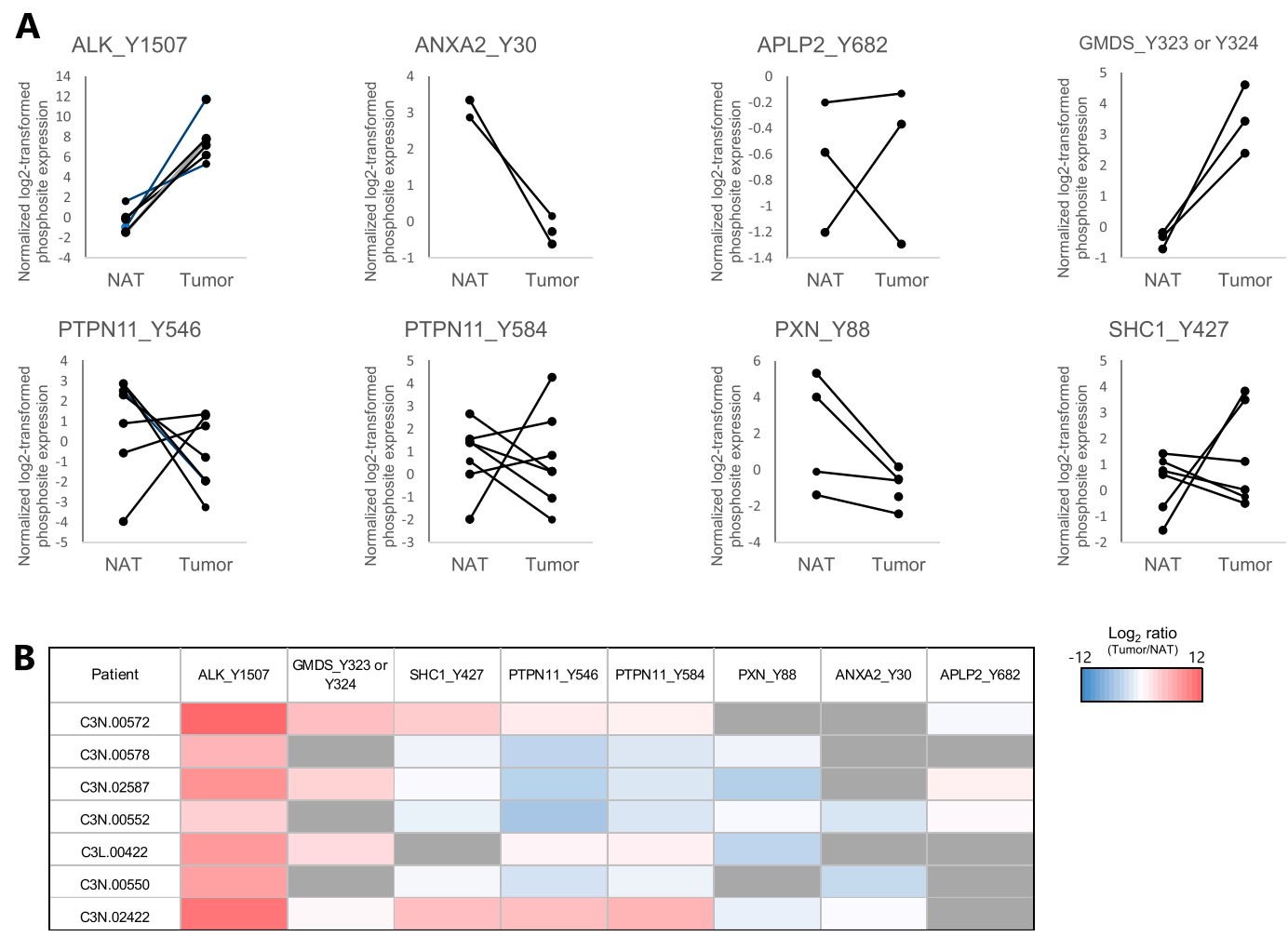

**Figure 7. Comparison of the phosphorylation levels of anaplastic lymphoma kinase (ALK) substrate candidates in tumour tissues and NATs of ALK-positive NSCLC patients.**
**(A)** Log$_2$-fold changes in the phosphorylation sites of ALK substrate candidates reported in a previous study (Gillette et al, 2020). **(B)** Heat map of log$_2$-fold changes in the phosphorylation sites of the ALK substrate candidates shown in Fig 7A.

Members of the cytoplasmic FMR1-interacting protein family (CYFIP1 and CYFIP2) are interactors of fragile X mental retardation protein (FMRP), a mRNA-binding protein that plays a key role in the translational silencing of its target mRNA (Schenck et al, 2001). In addition, CYFIPs bind and inhibit the WAVE regulatory complex (WRC), preventing the promotion of actin cytoskeleton reorganization. Phosphorylation of CYFIP2 (Thr$^{1092}$) was reported to regulate dendritic spine density and neurite outgrowth by decreasing the interaction affinity for the WRC complex (Lee et al, 2017). In this study, we identified CYFIP1/CYFIP2 (Tyr1054/1078), whose functions are not yet known, as ALK substrates. ALK might regulate protein synthesis and cytoskeletal dynamics via CYFIP phosphorylation.

Flotillin-1 has been reported to interact with ALK and regulate its lysosomal degradation through endocytosis in neuroblastoma cells (Tomiyama et al, 2014). Although ALK-dependent phosphorylation was reported, the phosphorylation sites were not elucidated. Here, we revealed that Tyr216 and Tyr238 are phosphorylated by ALK. It is expected that this information will be used to further elucidate the mechanisms that regulate ALK expression levels.

We also identified FAF2 (UBXD8), UBXN6 (UBXD1), ESYT1, and ESYT2 as ALK substrate candidates. These proteins were previously identified as endoplasmic reticulum-associated degradation (ERAD) network components (Nagahama et al, 2009; Christianson et al, 2011). Tyrosine phosphorylation of ERAD proteins by ALK has not been reported, but our data suggest the involvement of ALK in ERAD. As mentioned above, the ALK substrates identified in this study have novel functions in addition to the previously known functions of ALK. This finding will contribute not only to improving the prediction of ALK kinase activity but also to elucidating the pathophysiology of diseases featuring ALK gene mutations.

Furthermore, we found a difference in the experimental results obtained by signal suppression by ALK inhibitors and those obtained by signal enhancement, such as ALK overexpression. Specifically, in experiments in which *EML4-ALK*–positive NSCLC cells were treated with ALK inhibitors, the phosphorylation of ALK itself, PTPN11, FAF2, and SHC1 was strongly inhibited by gilteritinib, whereas the phosphorylation of other sites was inhibited to a lesser degree (Fig S5). In contrast, phosphorylation of these sites was

significantly increased in HEK 293 cells with doxycycline-induced ALK overexpression and in ALK-active neuroblastoma NB-1 and/or KP-N-RT-BM-1 cells (Fig 3). Typically, a single phosphorylation site is phosphorylated by multiple kinases. The contribution of each kinase is different for each phosphorylation site. Therefore, we consider these data to suggest that when ALK inhibitors are added, phosphorylation at sites that are highly phosphorylated by ALK is inhibited, and phosphorylation at sites that are less frequently phosphorylated by ALK are inhibited to a lesser degree. When the kinase activity of ALK is increased, phosphorylation by ALK is likely to increase, resulting in enhanced levels of ALK substrate phosphorylation. In the case of cancer, enhanced phosphorylation of these sites due to the activation of upstream kinases might be involved in cancer growth and metastasis and are promising targets for cancer therapy. Thus, further validation using different cells and inhibitors will be important to elucidate the "weight" of each KSR.

We also found that our custom KSR dataset improved the prediction of ALK activity in *EML4-ALK*-positive NSCLC cells (Figs 6 and S5). All variants of the EML4-ALK fusion protein contain the entire intracellular tyrosine kinase domain of ALK, encoded by exons 20 through 29. Thus, unsurprisingly, the substrates of ALK and EML4-ALK overlap, and ALK kinase activity prediction can be applied to EML4-ALK-positive cells.

The application of kinase activity prediction to clinical samples is very important for precision cancer medicine. We used clinical phosphoproteome data from the CPTAC database to compare ALK substrates in tumour tissues and NATs of *EML4-ALK*–positive NSCLC patients (Fig 7) and found that the phosphorylation of ALK and GMDS was increased in all tumour tissues, but the phosphorylation of SHC1 and PTPN11 was decreased in four patients and increased in three patients. Because the CPTAC data were obtained from surgical specimens, ischaemia, and differences in conditions after resection but before cryopreservation may affect the phosphorylation status of the ALK substrates in these samples. Recently, a phosphoproteome analysis method using biopsy specimens that can be frozen immediately after resection was developed (Abe et al, 2020; Satpathy et al, 2020). State-of-the-art technologies such as this one will contribute to the acquisition of more accurate kinase activity profiles of cancer patients for optimal treatment selection and evaluation.

# Materials and Methods

## Establishment of cells with inducible ALK expression

To generate ALK expression vectors to achieve the Dox-inducible expression of C-terminal FLAG-tagged bait proteins, human ORFs, which are provided as pDONR223 vectors, were selected from Gateway-compatible human ORFeome collections, which are distributed by Addgene. For LR recombination, the in-house-designed destination vector pRetroX-TRE3G/FLAG/GW, which we obtained through ligation of the FLAG tag coding sequence and the Gateway recombination cassette into pRetroX-TRE3G (Clontech), was used. The constructed plasmids were packaged into retroviral particles by using the packaging cell line AmphoPack-293 (Clontech).

Retrovirus-containing medium was harvested, filtered and used for transduction of HEK 293 Tet-On 3G cells (Clontech). Cells were transduced with the pRetroX-Tet3G/FLAG/GW retroviral vector and were then selected with puromycin. The transduced cells were pooled. Inducible expression of ALK was achieved by the addition of Dox (50 ng/ml) to the medium.

## Cell lines and culture conditions

HEK 293 Tet-On 3G cells were cultured in Iscove's modified Dulbecco's medium high glucose (Nacalai Tesque) supplemented with 10% Tet System Approved FBS (Takara Bio) at 37°C in 10% $CO_2$. IMR-32, KP-N-RT-BM-1, and NB-1 human neuroblastoma cells were cultured in RPMI 1640 medium (Nacalai Tesque) supplemented with 10% FBS.

## Cell treatment and lysis

ALK expression was induced by the addition of Dox (50 ng/ml) for 0, 2, 4, 8, and 24 h, and cell pellets were lysed with lysis buffer (50 mM $NaHCO_3$, 12 mM sodium N-lauroyl sarcosinate, and 12 mM sodium deoxycholate) supplemented with cOmplete EDTA-free and PhosSTOP (Roche).

## Preparation of samples for MS-based proteome and phosphoproteome analysis

Each sample was boiled at 95°C for 5 min. Lysates were further sonicated with a Bioruptor sonicator (Cosmo Bio). Then, 2 mg of each sample was reduced with 10 mM TCEP, alkylated with 20 mM iodoacetamide, and quenched with 21 mM L-cysteine. The samples were digested with trypsin (protein weight ratio: 1/50; Wako) and Lys-C (1 mAU/25 μg; Wako) for 16 h at 37°C. Samples were acidified with 1% TFA and centrifuged at 20,000*g* for 10 min at 4°C. Supernatants were desalted, and 0.25% of each sample was used for tandem mass tag (TMT) labelling for proteome analysis. The remaining portions of the samples were subjected to phosphopeptide enrichment on an immobilized metal affinity chromatography (IMAC) column (Abe et al, 2020) and labelled with TMT 10plex reagent. Each proteome and phosphoproteome sample was subjected to TMT labelling according to the manufacturer's protocol. Then, 40% and 83.3% of labelled phosphopeptides from HEK 293 Tet-On 3G cells and neuroblastoma cells, respectively, were applied to enrich phosphotyrosine peptides. Phosphotyrosine enrichment was performed using a pY1000 antibody as previously reported (Abe et al, 2017b).

TMT-labelled peptides/phosphopeptides used for proteome and global phosphoproteome analysis were fractionated into seven fractions with $C^{18}$/SCX StageTips as described previously (Adachi et al, 2016).

## Preparation of samples for immunoprecipitation and immunoblotting

IMR-32 and NB-1 cells were lysed with cell lysis buffer containing 50 mM Tris–HCl (pH 7.5), 150 mM NaCl, 1 mM EDTA, 1% Triton X-100, 0.5% sodium deoxycholate, and 0.1% SDS and supplemented with

cOmplete EDTA-free and PhosSTOP. The lysates were sonicated with a Bioruptor sonicator and centrifuged at 4°C for 10 min at 14,000g.

For each reaction, 100 μl of a slurry of Protein-A magnetic beads (10515-1-AP; Bio-Rad) was washed three times with PBS-T. Four micrograms of LASP1 polyclonal antibody (10515-1-AP; Proteintech) was immobilized to the beads in TBS-T. The bead-antibody mixture was applied to the lysates and incubated at 4°C for 1 h. The antigen-bound beads were washed four times in wash buffer (TBS-T). After the final washing step, 40 μl of LDS sample buffer (NP0007; Invitrogen) containing 20 mM DTT was added to the beads. The samples were incubated at 70°C for 10 min. The eluates were analysed by immunoblotting as previously described (Abe et al, 2020).

For protein separation, a 5–20% XV Pantera gradient gel (DRC Tokyo) was used. Eluates were subjected to electrophoresis at 300 V for 15 min. Proteins were blotted to polyvinylidene difluoride (PVDF) membranes at 40 V for 70 min. The PVDF membranes were incubated with a primary antibody (LASP1 Polyclonal antibody [10515-1-AP; Proteintech]) and Phospho-Tyrosine (P-Tyr-1000) antibody (Cell Signalling Technology [# 8954]) overnight at RT and with a secondary antibody conjugated to HRP (18-8816; TrueBlot Anti-IgG HRP, Rockland) for 1 h. Chemiluminescent measurements of the blotted proteins were performed using ECL Blotting Reagents (RPN2109; Cytiva) with LAS 4000 (Cytiva).

### Preparation of samples for interactome analysis

HEK 293 Tet-On 3G cells cultured in 15-cm dishes were crosslinked with 0.25% (wt/vol) formaldehyde solution for 10 min. The cross-linking reaction was terminated by the addition of glycine at a final concentration of 0.25 M. Cells were washed twice with ice-cold PBS and lysed in cell lysis buffer (50 mM Tris–HCl (pH 7.5), 150 mM NaCl, 0.35% sodium lauroyl sarcosinate, 0.5% sodium deoxycholate, 0.1% sodium dodecyl sulfate, and cOmplete EDTA-free phosphatase inhibitor). Benzonase nuclease (25 U; Sigma-Aldrich) was then added to the lysate and incubated for 15 min at 37°C, and 3.3 mg of the lysate was incubated with 20 μl of anti-FLAG M2 agarose beads (Sigma-Aldrich) at 4°C for 2 h on a rotating shaker. The beads were washed twice with lysis buffer and three times with wash buffer (50 mM Tris–HCl [pH 7.5], 150 mM NaCl, 0.35% sodium lauroyl sarcosinate, and 0.5% sodium deoxycholate), and after a final wash, the complexes were eluted in 80 μl of a Flag peptide solution (500 μg/ml in wash buffer; Sigma-Aldrich). The samples were boiled for 30 min, reduced with 10 mM TCEP, alkylated with 20 mM iodoacetamide, and quenched with 21 mM L-cysteine. Samples were digested with trypsin and Lys-C for 16 h at 37°C. Samples were then acidified with 1% TFA and centrifuged at 20,000g for 10 min at 4°C. Supernatants were desalted with C$^{18}$ StageTips (Ishihama et al, 2006).

### In vitro kinase assay of the ALK protein and ALK substrates

Recombinant ALK, LYN, and MAP4K4 proteins were kindly provided by Carna Bioscience. ANXA2, APLP2, CNP, CYFIP2, EMD, ESYT1, ESYT2, FAF2, FLOT1, GMDS, LASP1, PIN4, PTPN11, PXN, PTPRS, PXN, SHC1, SMAD1, and UBXN6 were purchased from Origene. IRS was obtained from SAB. ATP (0 or 20 μM) was mixed with 8.3 μg/ml substrate dissolved in reaction buffer before incubation for 60 min at 30°C. After the kinase reaction, 10× PTS buffer was added and boiled at 95°C for 5 min. Then, the samples were reduced with 10 mM TCEP, alkylated with 20 mM iodoacetamide, and quenched with 21 mM L-cysteine before digestion with trypsin and Lys-C using the SP3 protocol (Hughes et al, 2019). In brief, 40 μl of a mixture of pre-washed Sera-Mag Speed Beads A and B (Thermo Fisher Scientific) and ethanol (EtOH) were added to each sample to a final concentration of 50% EtOH. Proteins were allowed to bind to the beads for 10 min, and the beads were then incubated for 2 min on a magnetic rack for immobilization. The supernatant was removed, and the beads were washed three times with 500 μl of 80% EtOH. The beads were resuspended in 100 μl of 50 mM ammonium bicarbonate (pH 8.0). Finally, 1 μg trypsin and 2 mAU Lys-C were added and incubated at 37°C overnight. Supernatants were acidified by adding 5 μl of 20% TFA and used for phosphopeptide enrichment via the StageTip-based IMAC method (Abe et al, 2020).

### LC–MS/MS analysis

Liquid chromatography-tandem mass spectrometry (LC–MS/MS) was performed with an UltiMate 3000 Nano LC system (Thermo Fisher Scientific) and an HTC-PAL autosampler (CTC Analytics) coupled to a Q Exactive, Q Exactive Plus or Orbitrap Fusion Lumos mass spectrometer (Thermo Fisher Scientific). For proteome and phosphoproteome analysis of HEK 293 Tet-On 3G cells, the nano-liquid chromatography gradient was composed of Buffer A (0.1% formic acid and 2% acetonitrile) with a gradient of 5–30% Buffer B (0.1% formic acid and 90% acetonitrile) over 145 min. The settings of the Q Exactive Plus mass spectrometer were similar to those described in a previous phosphoproteomic study (Abe et al, 2017a, 2020). For interactome analysis and the in vitro kinase assay, 85 and 30-min gradients from 5 to 30% Buffer B were used. The Q Exactive instrument was operated as previously described (Adachi et al, 2016). For proteome and phosphoproteome analysis of neuro-blastoma cells, a 145-min gradient from 5 to 30% solvent B (solvent A, 0.1% FA; solvent B, 0.1% FA, and 99.9% acetonitrile) was used. The settings of the Orbitrap Fusion Lumos mass spectrometer were essentially as described in our previous study.

### MS data analysis

Raw MS data were processed with MaxQuant (versions 1.5.1.2, 1.6.3.3, and 1.6.14.0 for HEK 293 Tet-On 3G cell, interactome, neuroblastoma cell, and in vitro kinase assay analyses, respectively) supported by the Andromeda search engine for peak detection and quantification (Cox & Mann, 2008). The MS/MS spectra were searched against the UniProt human database with the following search parameters: full tryptic specificity, up to two missed cleavage sites, carbamidomethylation of cysteine residues set as the fixed modification, and N-terminal protein acetylation and methionine oxidation set as variable modifications. For phosphoproteome analysis, phosphorylation of serine, threonine, and tyrosine was added as a variable modification. The search results were filtered to a maximum false discovery rate of 0.01 at the protein, peptide-spectrum match (PSM), and posttranslational modification (PTM) site levels.

We required two or more unique/razor peptides for protein identification and a ratio count of two or more for protein quantification. PTM sites with a measured localization probability >0.75 were considered to be localized. The MS/MS spectra used to identify the substrate candidates are shown in Fig S2.

## Bioinformatic analysis

Statistical analysis was carried out with Perseus 1.6.5.0 and 1.6.14.0 (https://maxquant.net/perseus/) (Tyanova et al, 2016). For the proteome and phosphoproteome data, the quantitative TMT reporter ion intensities were $\log_2$-transformed and normalized by the median centering of the values in each sample. Kinase activity prediction was performed using site-centric PTM-SEA and PTMsigDB v1.9.0 (Krug et al, 2019). PTM-SEA results were visualized with Coral (Metz et al, 2018). To assess sequence bias around the phosphorylation sites of ALK substrate candidates, sequence motif logo plots (amino acids between positions ±7 adjacent to the identified phosphorylated sites) were generated and visualized using iceLogo software (Colaert et al, 2009) with default parameters ($P < 0.01$). The identified phosphotyrosine sites were used as the common background. For interactome data, ALK interactors with statistically significant changes were identified by two-tailed Welch's $t$ test using the LFQ intensities of Dox (+) and Dox (–) samples. A permutation test was performed to calculate the adjusted $q$-values. Based on the fold change and $q$-value, significant differences for ALK interactors were determined as reported previously (Tusher et al, 2001). Modulated ALK interactors are summarized in the source data for the main figures. The protein association network based on ALK substrate candidates was obtained using the STRING database (version 11) (Szklarczyk et al, 2019). All active interaction sources were included in the network, and a medium confidence score of greater than 0.4 was needed.

## Data Availability

All raw data files generated in this study were deposited into jPOST, a public proteome database certified by the ProteomeXchange Consortium (Okuda et al, 2017), under accession number PXD027676/JPST001277, and PXD027677/JPST001278.

## Supplementary Information

## Acknowledgements

This study was supported in part by MEXT/JSPS KAKENHI grant numbers 16H06150, 19H03530 (both to J Adachi) and 20H03544 (to T Tomonaga) and by MHLW/AMED grant numbers JP21ck0106465h0003 and JP17ck0106170h0003 (both to J Adachi) and JP18ck0106231h0003 (to T Tomonaga).

## Author Contributions

J Adachi: conceptualization, data curation, formal analysis, funding acquisition, validation, visualization, project administration, and writing—original draft, review, and editing.
A Kakudo: formal analysis.
Y Takada: formal analysis.
J Isoyama: formal analysis.
N Ikemoto: formal analysis.
Y Abe: formal analysis.
R Narumi: formal analysis.
S Muraoka: formal analysis.
D Gunji: formal analysis.
Y Hara: formal analysis.
R Katayama: conceptualization.
T Tomonaga: conceptualization, funding acquisition, and writing—review and editing.

## Conflict of Interest Statement

The authors declare that they have no conflict of interest.

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
