## [Reviewer comments · Life Science Alliance]

Life Science Alliance

Systematic identification of ALK substrates by integrated phosphoproteome and interactome analysis

Jun Adachi, Akemi Kakudo, Yoko Takada, Junko Isoyama, Narumi Ikemoto, Yuichi Abe, Ryohei Narumi, Satoshi Muraoka, Daigo Gunji, Yasuhiro Hara, Ryohei Katayama, and Takeshi Tomonaga

DOI: <https://doi.org/10.26508/lsa.202101202>

Corresponding author(s): Jun Adachi, National Institute of Biomedical Innovation, Health and Nutrition

Review Timeline:

Submission Date:	2021-08-19
Editorial Decision:	2021-09-28
Revision Received:	2021-12-23
Editorial Decision:	2022-01-26
Revision Received:	2022-04-06
Editorial Decision:	2022-04-13
Revision Received:	2022-04-16
Accepted:	2022-04-19

Transaction Report:

September 28, 2021

Re: Life Science Alliance manuscript #LSA-2021-01202-T

Prof Jun Adachi
NIBIOHN
Laboratory of Proteome Research
Osaka
Japan

Dear Dr. Adachi,

Thank you for submitting your manuscript entitled "Systematic identification of ALK substrates by integrated phosphoproteome and interactome analysis" to Life Science Alliance. The manuscript was assessed by expert reviewers, whose comments are appended to this letter. We invite you to submit a revised manuscript addressing the Reviewer comments.

Thank you for this interesting contribution to Life Science Alliance. We are looking forward to receiving your revised manuscript.

Sincerely,

B. MANUSCRIPT ORGANIZATION AND FORMATTING:

Reviewer #1 (Comments to the Authors (Required)):

This manuscript seeks to define kinase-substrate relationships using interactome and phosphoproteome analysis. This is a particularly difficult problem as computational predictions depend on prior experimental data and experimental techniques are not easy or direct. Here, the authors use ALK as a prototype and use an inducible system to generate the data on phosphotyrosine phosphoproteome (655 sites) and interactome (formaldehyde crosslinking and label-free quantitation). Importantly, they overlapped the results to generate potential kinase substrate relationships (KSR). They performed a number of in vitro kinase experiments to validate the KSRs. Using a custom KSR database, they were able to predict the activity of several ALK inhibitors. In addition, they demonstrate that this additional information that they generated was useful to predict responses to a novel ALK inhibitor as well as in samples where the data was generated by the CPTAC consortium. The authors present solid data and their claims are justified based on the data. I think that this manuscript is suitable for publication after the following minor issues are addressed:

1. The authors should consider restating their findings in this sentence: "The protein expression level of ALK was increased 1.1- and 3.9-fold in KP-N-RT-BM-1 and NB-1 cells, respectively, compared to IMR-32 cells" as they might want to say that the levels of ALK were not increased in KP-N-RT-BM-1 cells as expected (as these cells have a mutation in ALK as opposed to amplification).
2. The authors should clarify in the following sentence that while KP-N-RT-BM-1 has an amplification, NB-1 cells have a mutation in ALK (as geneticists use a more narrow definition of "mutation")
"Compared to IMR-32 cells, the ALK-activated cell lines KP-N-RT-BM-1 and NB-1 have different ALK mutations but show a very similar pattern of kinome activity profiles"

Referee Cross-Comments:

Reviewer #2:

- Point 1: There are several reasons for ALK activation - one is overexpression through amplification so I think the current model should suffice at least for this study although expansion to signaling specifically initiated by other mechanisms would make it more comprehensive (though not take the current study closer to their goal)
- Point 2: Yes, the authors could attempt anti-pTyr blots as suggested although the current data is site-specific and the blots, even if they correlate, will not provide such information.
- Point 3: The data provided by authors in Figure 5 is from CPTAC studies and not the authors' data. They have provided an explanation for why the correlation might be low.

Reviewer #3:

I would presume that overexpression will cause activation and the interactome (at least a subset) contains a subset of "true and regulated" interactors.

Reviewer #2 (Comments to the Authors (Required)):

Activated versions of anaplastic lymphoma kinase (ALK) are important oncogenic drivers in different cancer entities, notably neuroblastoma and non-small cell lung carcinoma. Consequently, ALK tyrosine kinase inhibitors have been developed and are being clinically used for cancer treatment. Physiologically, ALK is important for aspects of neurogenesis. Knowledge of the full spectrum of ALK substrates is therefore of both general and medical interest. Adachi et al. have assessed this issue with a proteomic/phosphoproteomic approach. Using inducible overexpression of WT, FLAG-tagged full-length ALK in HEK293 cells, the alterations in the phosphoprotein pattern were determined by quantitative MS. Additionally, ALK interacting proteins were evaluated using formaldehyde-based protein crosslinking followed by ALK immunoprecipitation. Comparison of identified proteins revealed ALK associating phosphoproteins, of which several had not been previously identified. In vitro ALK kinase

assays with corresponding recombinant proteins supported that these proteins are candidate substrates. The identified phosphosites were used to score ALK activity in human cell lines harboring activated ALK versions and its perturbation by treatment with ALK inhibitors using PTM signature enrichment analysis (PTM-SEA). The authors demonstrated that inclusion of the newly identified phosphosites improved prediction of ALK activity and inhibitor response as compared with assessment based on the pattern of previously known phosphosites. Several newly identified phosphosites are discussed with respect to possible cellular functions, like regulation of TGFbeta signaling, regulation of cytoskeletal organization, or ERAD. The authors state that the data expand the repository of ALK kinase-substrate relationships (KSR), may be helpful in elucidating further cellular functions of ALK, and aid in assessment of ALK activity in tumor samples in absence and presence of ALK inhibitors.

General:

This is a careful and well-presented study, which in my opinion lacks however, independent confirmation and deeper analysis of at least a part of the identified phosphosites. Moreover, I have concerns with respect to the relevance of the applied methodology.

Points:

1. The authors successfully identified novel phosphosites based on overexpression of WT ALK in HEK293 cells. The underlying assumption that WT ALK overexpression is functionally equivalent to ligand-activation of the kinase or activation by mutation is probably not correct. Given the importance of ALK activity as oncogenic driver, the use of an activated ALK version and of a physiologically/pathologically more relevant cellular background would have been desirable (see also point 3). The crosslinking approach to identify ALK associated proteins used ALK-immunoprecipitates of induced vs. non-induced cells for comparison. While this should be correct for non-specifically associated proteins to antibodies/beads, IPs from induced but kinase-inhibitor treated cells would have been a more relevant control for identifying KSRs.
2. Independent confirmation of the occurrence of some newly identified phosphorylations in the context of endogenous, activated ALK in the neuroblastoma cell lines by IP/western blotting would strongly improve the study. This should be possible even without the development of novel tools. For example, IPs of SMAD1, MAP4K4, APLP2, or ESYT1 with subsequent anti-pY blots with or without prior inhibitor (or even better siRNA) perturbation appear feasible.
3. Assessment of inhibitor perturbation of identified phosphosites in EML4-ALK-positive NSCLC cells (Expanded Fig. 3) indicates that apart from the known substrates ALK itself, PTPN11, and SHC1, phosphorylation of the other protein sites is in fact NOT responding to gilteritinib, casting some doubt that these proteins are indeed relevant KSRs at least for ALK fusion proteins. Admittedly, analysis in tissue samples is more difficult. Still, data in Fig. 5 show clear upregulation of the ALK autophosphorylation in tumor tissue, while phosphorylation of the other sites mostly does not correlate. Again, this finding questions the usefulness of the newly identified sites for analyses with clinical relevance.

Reviewer #3 (Comments to the Authors (Required)):

The manuscript by Adachi et al. entitled "Systematic identification of ALK substrates by integrated phosphoproteome and interactome analysis" describes a combined mass spectrometry approach to identify 'true' ALK substrates and their modification sites. Although there are several approaches reported in the field of kinase-substrate relationship analysis, many in vitro result showed false positives since cellular location of substrates was not considered. Current study by the authors has employed to analyze interactome of ALK to reduce this false identification. The authors employed a combined biochemical and genetic approach by establishing doxycycline-induced ALK-overexpressing HEK-293 cells. Overall, the study was designed well and technically sound. I have one comment: Since ALK is one of receptor tyrosine kinases, would you be able to activate the ALK to find 'regulated and true' interactome and their tyrosine phosphorylation sites?

Reviewer #1 (Comments to the Authors (Required)):

> *This manuscript seeks to define kinase-substrate relationships using interactome and phosphoproteome analysis. This is a particularly difficult problem as computational predictions depend on prior experimental data and experimental techniques are not easy or direct. Here, the authors use ALK as a prototype and use an inducible system to generate the data on phosphotyrosine phosphoproteome (655 sites) and interactome (formaldehyde crosslinking and label-free quantitation). Importantly, they overlapped the results to generate potential kinase substrate relationships (KSR). They performed a number of in vitro kinase experiments to validate the KSRs. Using a custom KSR database, they were able to predict the activity of several ALK inhibitors. In addition, they demonstrate that this additional information that they generated was useful to predict responses to a novel ALK inhibitor as well as in samples where the data was generated by the CPTAC consortium. The authors present solid data and their claims are justified based on the data. I think that this manuscript is suitable for publication after the following minor issues are addressed:*

1. The authors should consider restating their findings in this sentence: "The protein expression level of ALK was increased 1.1- and 3.9-fold in KP-N-RT-BM-1 and NB-1 cells, respectively, compared to IMR-32 cells" as they might want to say that the levels of ALK were not increased in KP-N-RT-BM-1 cells as expected (as these cells have a mutation in ALK as opposed to amplification).

Suggested changes have been implemented as below. “The protein expression level of ALK was not increased in KP-N-RT-BM-1 cell, whereas increased 3.9-fold in NB-1 cells compared to IMR-32 cells.”

2. *The authors should clarify in the following sentence that while KP-N-RT-BM-1 has an amplification, NB-1 cells have a mutation in ALK (as geneticists use a more narrow definition of "mutation")*

"Compared to IMR-32 cells, the ALK-activated cell lines KP-N-RT-BM-1 and NB-1 have different ALK mutations but show a very similar pattern of kinome activity profiles"

According to the suggestion, we changed the sentence as follows.

“Compared to IMR-32 cells, KP-N-RT-BM-1 and NB-1 cells show a very similar pattern of kinome activity profile, although ALK is activated by the mutation and the amplification, respectively.”

Referee Cross-Comments:

Reviewer #2:

Point 1: There are several reasons for ALK activation - one is overexpression through amplification so I think the current model should suffice at least for this study although expansion to signaling specifically initiated by other mechanisms would make it more comprehensive (though not take the current study closer to their goal)

Point 2: Yes, the authors could attempt anti-pTyr blots as suggested although the current data is site-specific and the blots, even if they correlate, will not provide such information.

Point 3: The data provided by authors in Figure 5 is from CPTAC studies and not the authors' data. They have provided an explanation for why the correlation might be low.

Reviewer #3:

I would presume that overexpression will cause activation and the interactome (at least a subset) contains a subset of "true and regulated" interactors.

I totally agree with the referee cross-comments.

Reviewer #2 (Comments to the Authors (Required)):

Activated versions of anaplastic lymphoma kinase (ALK) are important oncogenic drivers in different cancer entities, notably neuroblastoma and non-small cell lung carcinoma. Consequently, ALK tyrosine kinase inhibitors have been developed and are being clinically used for cancer treatment. Physiologically, ALK is important for aspects of neurogenesis. Knowledge of the full spectrum of ALK substrates is therefore of both general and medical interest. Adachi et al. have assessed this issue with a proteomic/phosphoproteomic approach. Using inducible overexpression of WT, FLAG-tagged full-length ALK in HEK293 cells, the alterations in the phosphoprotein pattern were determined by quantitative MS. Additionally, ALK interacting proteins were evaluated using formaldehyde-based protein crosslinking followed by ALK immunoprecipitation. Comparison of identified proteins revealed ALK associating phosphoproteins, of which several had not been previously identified. In vitro ALK kinase assays with corresponding recombinant proteins supported that these proteins are candidate substrates. The identified phosphosites were used to score ALK activity in human cell lines harboring activated ALK versions and its perturbation by treatment with ALK inhibitors using PTM signature enrichment analysis (PTM-SEA). The authors demonstrated that inclusion of the newly identified phosphosites improved prediction of ALK activity and inhibitor response as compared with assessment based on the pattern of previously known phosphosites. Several newly identified phosphosites are discussed with respect to possible cellular functions, like regulation of TGFbeta signaling, regulation of cytoskeletal organization, or ERAD. The authors state that the data expand the repository of ALK kinase-substrate relationships (KSR), may be helpful in elucidating further cellular functions of ALK, and aid in assessment of ALK activity in tumor samples in absence and presence of ALK inhibitors.

General:

This is a careful and well-presented study, which in my opinion lacks however, independent confirmation and deeper analysis of at least a part of the identified phosphosites. Moreover, I have concerns with respect to the relevance of the applied methodology.

Points:

1. The authors successfully identified novel phosphosites based on overexpression of WT ALK in HEK293 cells. The underlying assumption that WT ALK overexpression is functionally equivalent to ligand-activation of the kinase or activation by mutation is probably not correct. Given the importance of ALK activity as oncogenic driver, the use of an activated ALK version and of a physiologically/pathologically more relevant cellular background would have been

desirable (see also point 3). The crosslinking approach to identify ALK associated proteins used ALK-immunoprecipitates of induced vs. non-induced cells for comparison. While this should correct for non-specifically associated proteins to antibodies/beads, IPs from induced but kinase-inhibitor treated cells would have been a more relevant control for identifying KSRs.

As the reviewer pointed out, the difference between ALK signaling by overexpression and mutation and whether it is appropriate to use the results obtained from the analysis of HEK293 cells for ALK signaling analysis in cancer are very important points. Therefore, we performed phosphoproteomic analysis of not only ALK overexpressed cell but activated ALK-mutant neuroblastoma cells. As shown in Fig. 3C, we compared the kinase activity profiles of ALK-mutant and overexpressed neuroblastoma cells (KP-N-RT-BM-1 and NB-1 cells) and found that the profiles of the two cells were similar, although not completely identical. Furthermore, using the substrate information obtained from Hek293 cells, Figure 3 and Figure 4 show that the activation of ALK and the suppression of ALK activity by inhibitors can be more clearly captured using neuroblastoma and lung cancer cells, respectively. Of course, difference of cellular background will affect ALK signaling including some substrates, but we believe that an important result of our study is that we were able to show that there is a common substrate candidate between HEK293 cells and neuroblastoma and lung cancer cells, which contributes to improving the accuracy of kinase activity profiling.

2. Independent confirmation of the occurrence of some newly identified phosphorylations in the context of endogenous, activated ALK in the neuroblastoma cell lines by IP/western blotting would strongly improve the study. This should be possible even without the development of novel tools. For example, IPs of SMAD1, MAP4K4, APLP2, or ESYT1 with subsequent anti-pY blots with or without prior inhibitor (or even better siRNA) perturbation appear feasible.

We do not believe that the Anti pY blot can be used for validation as it does not provide information on the levels of individual phosphorylation sites and therefore, we cannot conclude whether it correlates with our phosphoproteome data. Instead, we added MS/MS spectra used for the identification of each phosphorylation site to the Supplemental Figure S4.

3. Assessment of inhibitor perturbation of identified phosphosites in EML4-ALK-positive NSCLC cells (Expanded Fig. 3) indicates that apart from the known substrates ALK itself, PTPN11, and SHC1, phosphorylation of the other protein sites is in fact NOT responding to gilteritinib, casting some doubt that these proteins are indeed relevant KSRs at least for ALK fusion proteins. Admittedly, analysis in tissue samples is more difficult. Still, data in Fig. 5 show

clear upregulation of the ALK autophosphorylation in tumor tissue, while phosphorylation of the other sites mostly does not correlate. Again, this finding questions the usefulness of the newly identified sites for analyses with clinical relevance.

The reviewer's point about phosphorylation changes when inhibitors are added to EML4-ALK positive NSCLC cells is important. The phosphorylation of ALK itself, PTPN11, FAF2 and SHC1 is strongly inhibited by gilteritinib, whereas the other sites are less inhibited. Typically, a single phosphorylation site is phosphorylated by multiple kinases. The contribution of each kinase is different for each phosphorylation site. Therefore, our interpretation is that when ALK inhibitors are added, the phosphorylation sites with high ALK contribution are inhibited and the phosphorylation sites with low contribution are less inhibited. This difference in contribution may provide useful information for weighting edges in the network analysis of kinase signalling in the future.

Regarding the CPTAC data, only a small number of patients showed increased phosphorylation in tumours, even with the known substrates PTPN11 and SHC1. This result suggests the limitations of using surgical specimens for phosphorylation analysis, as they cannot be frozen immediately and are subject to haemostasis. Phosphorylation status in surgical tissues is largely affected by ischemia during surgery and time lag until sample collection after cancer tissue resection, which might cause the differences in the phosphorylation sites between our results and CPTAC data. As noted in the Discussion, phosphorylation analysis of biopsy specimens frozen immediately after collection may reveal the usefulness of the ALK substrate candidates identified in this study with clinical relevance.

Reviewer #3 (Comments to the Authors (Required)):

The manuscript by Adachi et al. entitled "Systematic identification of ALK substrates by integrated phosphoproteome and interactome analysis" describes a combined mass spectrometry approach to identify 'true' ALK substrates and their modification sites. Although there are several approaches reported in the field of kinase-substrate relationship analysis, many in vitro result showed false positives since cellular location of substrates was not considered. Current study by the authors has employed to analyze interactome of ALK to reduce this false identification. The authors employed a combined biochemical and genetic approach by establishing doxycycline-induced ALK-overexpressing HEK-293 cells. Overall, the study was designed well and technically sound. I have one comment: Since ALK is one of receptor tyrosine kinases, would you be able to activate the ALK to find 'regulated and true' interactome and their tyrosine phosphorylation sites?

We thank the reviewer for the positive comments and have addressed his or her comment. Downstream analysis of ligand activated ALK was reported in EMBO journal this year (Borenäs et. al. The EMBO J. (2021) 40:e105784). Their data clearly shows ALKAL2, an ALK ligand, activates ALK-downstream signaling (We added this paper as a reference, line 206). Thus, it is possible to perform phosphotyrosine analysis. However, anti-ALK antibodies that capture formaldehyde (FA)-treated endogenous ALK have not been obtained, and thus interactome analysis could not be performed in our lab. In contrast, our method allows overexpression of FLAG-tagged ALK, followed by cross-linking with FA, and efficient enrichment of ALK complexes using anti-FLAG antibodies. This enabled us to perform interactome analysis.

January 26, 2022

Re: Life Science Alliance manuscript #LSA-2021-01202-TR

Prof. Jun Adachi
National Institute of Biomedical Innovation, Health and Nutrition
Laboratory of Proteome Research
7-6-8 Saito-Asagi
Ibaraki, Osaka 567-0085
Japan

Dear Dr. Adachi,

Thank you for submitting your revised manuscript entitled "Systematic identification of ALK substrates by integrated phosphoproteome and interactome analysis" to Life Science Alliance. The manuscript has been seen by an original reviewer whose comments are appended below. While the reviewer continues to be overall positive about the work in terms of its suitability for Life Science Alliance, some important issues remain.

Our general policy is that papers are considered through only one revision cycle; however, we are open to one additional short round of revision. Please note that I will expect to make a final decision without additional reviewer input upon resubmission.

We agree that point #2 needs to be clearly discussed in the manuscript - if some of these sites have little or no evidence they are even regulated by ALK, this is a concern. Also, the phospho-antibody experiment as corroborating evidence is reasonable.

Please submit the final revision within two months, along with a letter that includes a point by point response to the remaining reviewer comments.

To upload the revised version of your manuscript, please log in to your account: <https://lsa.msubmit.net/cgi-bin/main.plex>
You will be guided to complete the submission of your revised manuscript and to fill in all necessary information.

B. MANUSCRIPT ORGANIZATION AND FORMATTING:

Sincerely,

Reviewer #2 (Comments to the Authors (Required)):

Adachi et al. have provided responses to my concerns with the previous manuscript in their rebuttal letter. There are also responses by the other reviewers to my critics, which I gratefully acknowledge.

Related to my suggestion of assessing tyrosine phosphorylation of at least one of the newly identified putative ALK substrates by an alternative technique such as IP and immunoblotting with anti-pY antibodies, the authors have added MS/MS data to substantiate their identification of phosphosites. Otherwise, the manuscript was not altered to better reflect somehow the issues raised in my review.

After all, it is the responsibility of the authors and the editor what they wish to publish, and if they consider independent suggestions for improvement or not.

Still, I would like to make two points including suggesting at least some text amendments:

1. Despite the arguments of the authors and the co-reviewers, I believe that confirmation of newly identified phosphosites by an independent method would certainly improve the picture of ALK signaling in the investigated cells. Obviously, anti-PY antibodies will not provide site information, but would indicate if the (elaborately discussed) newly identified candidate substrates are tyrosine phosphorylated at all to a potentially meaningful level. I found this a doable experiment. Site-specific antibodies would obviously provide even better independent information. Such assays may also be a suitable way to continue towards clinical applications.

A) I suggest the authors consider this in their future work, and B) make a statement in the Discussion that such experiments are desirable.

2. Related to my concern that only some of the phosphosites respond to ALK inhibition, the authors state in the rebuttal letter: "The reviewer's point about phosphorylation changes when inhibitors are added to EML4 ALK positive NSCLC cells is important. The phosphorylation of ALK itself, PTPN11, FAF2 and SHC1 is strongly inhibited by gilteritinib, whereas the other sites are less inhibited. Typically, a single phosphorylation site is phosphorylated by multiple kinases. The contribution of each kinase is different for each phosphorylation site. Therefore, our interpretation is that when ALK inhibitors are added, the phosphorylation sites with high ALK contribution are inhibited and the phosphorylation sites with low contribution are less inhibited." In other words, the sites, which are not inhibited, are unlikely to be ALK substrate sites and are not even indirectly (through another kinase) dependent on ALK activity? This was exactly my concern.

I believe this issue requires some treatment with new text in the Discussion.

Reviewer #2 (Comments to the Authors (Required)):

Adachi et al. have provided responses to my concerns with the previous manuscript in their rebuttal letter. There are also responses by the other reviewers to my critics, which I gratefully acknowledge.

Related to my suggestion of assessing tyrosine phosphorylation of at least one of the newly identified putative ALK substrates by an alternative technique such as IP and immunoblotting with anti-pY antibodies, the authors have added MS/MS data to substantiate their identification of phosphosites. Otherwise, the manuscript was not altered to better reflect somehow the issues raised in my review.

After all, it is the responsibility of the authors and the editor what they wish to publish, and if they consider independent suggestions for improvement or not.

Still, I would like to make two points including suggesting at least some text amendments:

1. Despite the arguments of the authors and the co-reviewers, I believe that confirmation of newly identified phosphosites by an independent method would certainly improve the picture of ALK signaling in the investigated cells. Obviously, anti-PY antibodies will not provide site information, but would indicate if the (elaborately discussed) newly identified candidate substrates are tyrosine phosphorylated at all to a potentially meaningful level. I found this a doable experiment. Site-specific antibodies would obviously provide even better independent information. Such assays may also be a suitable way to continue towards clinical applications.

A) I suggest the authors consider this in their future work, and B) make a statement in the

Discussion that such experiments are desirable.

As the reviewer suggested, we performed an IP-Western blotting experiment targeting LASP1 as shown in Supplemental Figure S3. We also describe the importance of validating our results as follows.

line 217-222

Phosphorylation sites of ALK substrate candidates were identified based on the MS/MS spectrum (Supplemental Figure S2). In the future, this result should be confirmed by other methods, such as western blotting using a site-specific antibody or a combination of immunoprecipitation and immunoblotting using an anti-phosphotyrosine antibody, as shown in Supplemental Figure S3.

2. Related to my concern that only some of the phosphosites respond to ALK inhibition, the authors state in the rebuttal letter: "The reviewer's point about phosphorylation changes when inhibitors are added to EML4 ALK positive NSCLC cells is important. The phosphorylation of ALK itself, PTPN11, FAF2 and SHC1 is strongly inhibited by gilteritinib, whereas the other sites are less inhibited. Typically, a single phosphorylation site is phosphorylated by multiple kinases. The contribution of each kinase is different for each phosphorylation site. Therefore, our interpretation is that when ALK inhibitors are added, the phosphorylation sites with high ALK contribution are inhibited and the phosphorylation sites with low contribution are less inhibited."

In other words, the sites, which are not inhibited, are unlikely to be ALK substrate sites and are not even indirectly (through another kinase) dependent on ALK activity? This was exactly my concern.

I believe this issue requires some treatment with new text in the Discussion.

As the reviewer pointed out, it is possible that other kinases phosphorylate the substrates of ALK in vivo. I think it is important that the relationship between a kinase and a substrate is not "all or none". I believe that the weight of the relationship between a kinase and a substrate is variable and depends on the cell type and cellular environment. When the weight is light, it is not surprising that the phosphorylation state of the substrate is unlikely to be inhibited. If the kinase is activated by overexpression or other means and the weight becomes heavy, it is quite possible that the phosphorylation of the substrate will be increased.

We added new text to the Discussion (lines 263-282) as follows:

Furthermore, we found a difference in the experimental results obtained by signal suppression by ALK inhibitors and those obtained by signal enhancement, such as ALK overexpression. Specifically, in experiments in which EML4-ALK-positive NSCLC cells were treated with ALK inhibitors, the phosphorylation of ALK itself, PTPN11, FAF2, and SHC1 was strongly inhibited by gilteritinib, whereas the phosphorylation of other sites was inhibited to a lesser degree (Supplemental Figure S5). In contrast, phosphorylation of these sites was significantly increased in HEK-293 cells with doxycycline-induced ALK overexpression and in ALK-active neuroblastoma NB-1 and/or KP-N-RT-BM-1 cells (Figure 3). Typically, a single phosphorylation site is phosphorylated by multiple kinases. The contribution of each kinase is different for each phosphorylation site. Therefore, we consider these data to suggest that when ALK inhibitors are added, phosphorylation at sites that are highly phosphorylated by ALK is inhibited, and phosphorylation at sites that are less frequently phosphorylated by ALK are inhibited to a lesser degree. When the kinase activity of ALK is increased, phosphorylation by ALK is likely to increase, resulting in enhanced levels of ALK substrate phosphorylation. In the case of cancer, enhanced phosphorylation of these sites due to the activation of upstream kinases might be involved in cancer growth and metastasis and are promising targets for cancer therapy. Thus, further validation using different cells and inhibitors will be important in order to elucidate the “weight” of each kinase-substrate relationship.

April 13, 2022

RE: Life Science Alliance Manuscript #LSA-2021-01202-TRR

Prof. Jun Adachi
National Institute of Biomedical Innovation, Health and Nutrition
Laboratory of Proteome Research
7-6-8 Saito-Asagi
Ibaraki, Osaka 567-0085
Japan

Dear Dr. Adachi,

Thank you for submitting your revised manuscript entitled "Systematic identification of ALK substrates by integrated phosphoproteome and interactome analysis". We would be happy to publish your paper in Life Science Alliance pending final revisions necessary to meet our formatting guidelines.

- please upload your main and supplementary figures as single files
- please add the Twitter handle of your host institute/organization as well as your own or/and one of the authors in our system
- please make sure the author order in your manuscript and our system match and all contributing authors are added to our system
- please upload your main figures as single files; these will be displayed in-line in the HTML version of your paper, so please provide them as single page files (Figures 3 & 4 currently span 2 pages); we do not have a limit on the number of main figures and these can be split if necessary for space
- please be sure that the contribution of all authors is inserted in both the manuscript text and the system

A. FINAL FILES:

B. MANUSCRIPT ORGANIZATION AND FORMATTING:

Sincerely,

April 19, 2022

RE: Life Science Alliance Manuscript #LSA-2021-01202-TRRR

Prof. Jun Adachi
National Institute of Biomedical Innovation, Health and Nutrition
Laboratory of Proteome Research
7-6-8 Saito-Asagi
Ibaraki, Osaka 567-0085
Japan

Dear Dr. Adachi,

Thank you for submitting your Resource entitled "Systematic identification of ALK substrates by integrated phosphoproteome and interactome analysis". It is a pleasure to let you know that your manuscript is now accepted for publication in Life Science Alliance. Congratulations on this interesting work.

DISTRIBUTION OF MATERIALS:

Again, congratulations on a very nice paper. I hope you found the review process to be constructive and are pleased with how the manuscript was handled editorially. We look forward to future exciting submissions from your lab.

Sincerely,
